# Design and Assembly of a Miniature Catheter Imaging System for In Vivo Heart Endoscopic Imaging

**DOI:** 10.3390/s24196216

**Published:** 2024-09-25

**Authors:** Walter Messina, Lorenzo Niemitz, Simon Sorensen, Claire O’Dowling, Piotr Buszman, Stefan Andersson-Engels, Ray Burke

**Affiliations:** 1Biophotonics Group, Tyndall National Institute, University College Cork, T12 R5CP Cork, Ireland; lorenzo.niemitz@tyndall.ie (L.N.); simon.sorensen@tyndall.ie (S.S.); claire.odowling@tyndall.ie (C.O.); stefan.andersson-engels@tyndall.ie (S.A.-E.); 2Centre for Research in Vascular Biology, APC Microbiome Ireland, University College Cork, T12 K8AF Cork, Ireland; 3Centre for Cardiovascular Research and Development of American Heart of Poland, 40-028 Katowice, Poland; piotr.buszman@ahop.pl; 4School of Physics, University College Cork, T12 K8AF Cork, Ireland

**Keywords:** microcamera, cardiovascular imaging, biophotonics, catheterisation, prototyping

## Abstract

In this paper, we present the design and fabrication of a novel chip-on-tip catheter, which uses a microcamera and optical fibres to capture in vivo images in a beating porcine heart thanks to a saline flush to clear the blood field. Here, we demonstrate the medical utility and mechanical robustness of this catheter platform system, which could be used for other optical diagnostic techniques, surgical guidance, and clinical navigation. We also discuss some of the challenges and system requirements associated with developing a miniature prototype for such a study and present assembly instructions. Methods of clearing the blood field are discussed, including an integrated flush channel at the distal end. This permits the capture of images of the endocardial walls. The device was navigated under fluoroscopic guiding, through a guiding catheter to various locations of the heart, where images were successfully acquired. Images were captured at the intra-atrial septum, in the left atrium after a trans-septal cross procedure, and in the left ventricle, which are, to the best of our knowledge, the first images captured in an in vivo beating heart using endoscopic techniques.

## 1. Introduction

Since the early days of the 19th century with the invention of endoscopes, clinicians across different medical fields have relied upon images captured directly inside the patient for accurate diagnostics and treatment. During the last decades, there has been a constant drive to reduce the size of imaging devices for medical use, and endoscopy techniques are now routinely used for accurate diagnosis and minimally invasive surgeries (MIS) [1,2,3,4]. Increasingly smaller and more flexible [5] devices mean a reduced negative impact on the patient upon introduction, allowing the clinicians to reach sites and very small cavities that were previously inaccessible [6] Laparoscopes, fiberscopes, and microcameras mounted on flexible catheters have gained widespread use in multiple fields including otolaryngology [7,8], ophthalmology [9], gastroenterology [10,11], bronchoscopy [12], urology [13], angioscopy, cardioscopy [14,15,16], and arthroscopy [17,18]. Due to their small size, these surgical tools enable MIS, diagnosis, procedure evaluation and monitoring, and can further be combined with a biopsy tool [19]. Additional optical and ultrasound techniques have been integrated onto surgical tools for improved detection efficiency, including spectrophotometry [20], fluorescence imaging [21], optical coherence tomography (OCT) [22,23], and intravascular ultrasound (IVUS) [24]. There are several companies that offer microcamera optical sensors for endoscopic applications for different medical fields. Among these, the most popular ones are AMS OSRAM and OmniVision. Many other companies have packaged them in full systems ready to be used by clinicians. The smallest camera systems available on the market have a range of resolution from 200 × 200 pixels to 720 × 720 pixels and they have a footprint ranging from 0.65 mm x0.65 mm to 1.00 mm × 1.00 mm. Integrated systems including illumination can have an overall diameter around 2–3 mm and they are usually equipped with a simple white LED light, installed directly at the distal end (on systems with larger footprint) or carried at the camera location thanks to optical fibres [8,9,25,26,27]. Despite these efforts, so far, no imaging systems designed for sophisticated clinical diagnostic procedures in compact spaces have been presented.

## 2. Study Aim

In this paper, we present and discuss methodologies for designing and fabricating a novel chip-on-tip catheter that uses a microcamera, optical fibres for illumination, and a flush/irrigation system to clear the field of view of blood for in vivo intracardiac applications. To the best of our knowledge, there is no other paper describing a relatively simple system capable of capturing video images of a beating heart in vivo. We believe that this system could be further expanded and applied to many different other sensing parameters and several different types of sensors and biosensors.

Cardiac catheterisation is a very common procedure, and micro-image sensors such as the AMS NanEye and Omnivision sensors may prove valuable tools for cardiovascular imaging through a guiding catheter and the execution of a trans-septal cross. Small-scale integration of such a compact device requires robust prototype design and assembly processes, which were developed as part of this work. These processes are adaptable to different small footprint applications beyond cardiovascular imaging, as discussed in this paper. Two in vivo studies were conducted on a porcine animal model with the goal of assessing acute performance characteristics of the intra-vascular camera, to assess the ease and convenience with which it is delivered to the various treatment sites, and to test the system robustness during the procedure.

## 3. Imaging System Design

For our system, we used a NanEye microcamera (AMS (Premstaetten, Austria) OSRAM (Premstaetten, Austria), NE2D RGB V120F2.8 2m, field of view 120°, F 2.8). This camera was selected for its small footprint combined with image quality. The body of the imaging sensor is a parallelepiped that has nominal dimensions of 1.10 mm × 1.10 mm × 1.50 mm. There are four thin shielded electrical wires going from the back of the camera to the proximal end of the catheter system in a flat edge connector for powering and communicating with the interface board (BAPIS AC62KUSB, Ergolding, Germany). Illumination at the distal end of the catheter, at the camera’s face, is provided by two 200 µm diameter glass fibres (Thorlabs, FT200EMT (Bergkirchen, Germany)) flat polished to ensure maximum light transmission and finish. Fibre illumination allows for flexibility in wavelength delivery, small footprint, and eliminates heating concerns due to distally mounted LEDs. The presence of the two optical fibres creates a platform for other sensing modalities in addition to white-light imaging, such as diffuse reflectance spectroscopy (DRS). It also helps to deliver a better illumination profile, since the NA of the fibre is quite limited, and no lensing was added in these design iterations. The system itself is 2.00 m in length with a diameter of 2.1 mm to allow for navigation to take place. Navigation during cardiac catheterisation is performed under fluoroscopic guidance. To detect devices such as catheters, they are often fitted with a marker band which is visible on the X-ray image. The importance of radio-opacity of the system was highlighted by the collaborating clinicians, who found the design of the first-generation devices difficult to see during fluoroscopic imaging; hence, two 1.50 mm platinum–iridium wires were integrated behind the camera. The system is guided to location by pushing it through a guiding delivery catheter.

### 3.1. Alignment

The distal alignment piece, shown in Figure 1, serves during system assembly for component alignment in the x and y directions, as well as the z direction (along the length of the catheter). It also controls the distance between the fibres and other components for repeatability, ensures optical isolation of the camera sensor, and helps to provide a seal with the sheathing lumen. It consists of a custom 3D printed header piece (Form3, FormLabs; Clear Resin V4, FormLabs (Somerville, MA, USA)) that can accommodate the sensing, illumination, and flush channels. The printed parts were designed to be larger than the actual dimensions of the parts to account for shrinkage in the printing process and glueing.

A few experimental layouts were tested to identify the limitation of the 3D printing technology used and to investigate materials. One of the early designs also included the possibility of a prototype for a side-facing imaging system, which, however, was discarded for this study as side imaging was not required. An STL file (Appendix A) of the latest design is attached to the paper for the readers if needed (Figure 2).

### 3.2. Imaging in a Blood Field

In order to image within the cardiovascular system, the blood field needs to be cleared. Several methods were investigated to achieve this, most involving some form of saline solution flush. One of the biggest challenges lies in the continuous inflow of blood into the imaging area. Therefore, some isolation procedures need to be used to keep the cleared area free from blood. By far, the most effective method we found in this study was placing the guiding catheter against the tissue to act as a seal. An internal flush channel integrated into the device can introduce saline to clear the field of view. A separate flush can be introduced through the guiding catheter should more flow be required. An important feature of our imaging system is this high-volume flush at the distal end of the catheter. A 1.20 mm outer diameter (1.00 mm inner diameter) braided polyimide tubing (Nordson medical, Salem, NH, USA) was used to deliver the saline solution flush to clear the field of view in front of the camera. While compressed into an oval shape to fit into the distal end of the catheter, the proximal end of the tubing keeps its circular shape and is encapsulated, using biocompatible adhesive, into a female Luer that can be connected to syringes filled with saline solution. To the best of our knowledge, this is the first demonstration of a catheter integrated imaging system with a flush channel. As a test, a 3.00 mL syringe was completely emptied in less than 1 s by manual operation at full speed, demonstrating that this system can provide a fast and efficient clearance of the camera’s field of view.

Preliminary in vivo testing showed that the internal flush’s directional approach, as shown in Figure 3a, assisted in clearing the field of view in front of the sensor. The combined action of flushing with saline solution from the guiding catheter, as in Figure 3b, resulted in sufficient volume to easily clear the field of view. Most important, however, remained the isolation of the flushed area from inflowing blood using the guiding catheter.

## 4. Assembly Methods

The prototype device was assembled manually in house. Fixing is achieved using a USP Class IV biocompatible UV curing adhesive (NOA68, Norland Jamesburg, NJ, USA). The adhesive remains a liquid until cured and so allows for manipulation and precise alignment until it is cured. Temporary fixing can be achieved with only a small amount of the adhesive, which can be touched up later. Small amounts of the glue can be picked up and placed using a micro-applicator, which could be a sharp instrument such as a needle or, if scratching the optical components is a concern, a sharpened piece of wood, such as a whittled toothpick. An adjustable microscope arm also proves helpful.

The assembly process of the distal alignment piece is seen in Figure 4, which is described in detail below. The multi-step process is adaptable to different small-scale integration challenges and separates the assembly process into two parts, aligning all the components using the distal alignment piece and then marrying the lumen and the alignment piece along with proximal connectors to form a finished system. (a) The camera and optical fibres, and flush, are threaded through the sheathing by temporarily attaching them to a guidewire with a weak adhesive, which is pushed through the lumen. The camera should be inserted first as it is the most cumbersome and difficult part to push through, due to very flexible wires. The components now protrude through the front of the lumen. Here, they can be fixed to the distal alignment piece, in what is the first primary step of the assembly process. The fibres are cleaved prior to fixing, to ensure a good termination. The flush channel is compressed into an oval shape prior to fixing it into the holder end to ensure it fits into the assembly. (b) Prior to assembly, supports are removed from the 3D printed alignment part, and the part is post processed to remove excess material using thin files and needles. This ensures a smooth finish in the areas where parts are placed. The small dimensions lead to tight tolerances, and this step ensures all the parts fit the lumen. (c) The camera is now ready to be fitted into the square-shaped allotment. A thin layer of UV curable adhesive is deposited at the distal end of the 3D printed support on the three sides of the rectangular shape. The camera is fitted to ensure that no glue overflows in front of the lens and it is cured in place. If needed, a light clamping system can be used to keep the camera in place and flat and flush with the distal end of the support. Finally, a few drops of UV adhesive are added at the back of the camera to pot the camera with the support and strengthen the wire connection to the unit. During this step, the platinum–iridium wires for X-ray guidance can be also fitted behind the camera body and above its wires. (d) In order to fix the optical fibres, a very thin layer of UV glue is deposited at the distal end of the support dedicated to the optical fibres. The fibres are placed at the distal end of the support and cured, while attention is paid to make sure that they are aligned flat on the top of the camera and no glue is overflowing in front of them. A small clamp system is advisable to keep the fibres in place. Additional glue can be used at the tapered end of the support to provide additional strength to the bonding. It is also advisable to proceed with one fibre at a time. (e) The same procedure used for the fibres applies to the flushing system. However, being that the material is different from the fibres, it is advisable not to use UV glues but a cyanoacrylate-based glue instead. It has been noticed that the bonding using UV glues with polyimide is much weaker compared with other types of adhesives. Also, due to the front shape of the support, it is important to flatten the front part of the flush, making it become slightly oval. The tapered end of the support offers some space to place some additional glue to increase the bonding strength. (f) The outside of the assembled camera support heads is coated with a thin layer of glue. While press fitting the part inside the catheter, it is important to be careful of any overflowing glue. Should this get in front of any of the assembled parts, they must be cleaned immediately and the insertion of the part should be proceeded with slowly. If the catheter is UV transparent, this should be performed easily and, once happy with the position, the curing can start. Using alternative types of glues may be necessary in this step, although this could introduce timing difficulties.

The two optical fibres can be inserted into a single SMA connector to provide illumination. Should the sensing modality be required, they can be separated, at the manufacturing stage, into two individual connectors. For example, in DRS, one fibre can be used to provide broadband illumination and a second one can be used to collect diffused light for analysis with a spectrometer. Figure 5 shows images of the physical integration, back-end termination, and the final system.

## 5. Imaging Results

The assembled device was successfully constructed and shipped to the study clinic in the American Heart of Poland (Katowice, Poland) and the imaging of the endocardium at multiple locations was achieved on porcine animal models (ethical approval obtained from Localna Komisja Etyczna, Katowice, Poland, 41/20 and 58A/20). The device was navigated through the vascular system of the porcine animal with images captured in the right atrium, left atrium, and left ventricle of the animal’s heart. A sealing technique using the guiding catheter lightly pressed against the cardiac walls and saline solution flushing was used to clear the field of view in all cases. Best quality images were achieved when the camera device was slightly recessed inside the guiding catheter, the latter of which shields the device from additional blood flow after the field of view is flushed. This camera positioning can be seen well in Figure 6a–e, where the two marker wires embedded in the camera system are seen positioned inside of the guiding catheter, the tip of which has its own circular marker band. In the camera images, Figure 6b,d,f, the guiding catheter can also be seen as it creates an annular feature in the image. Part of the imaging field of view is occupied by the catheter, with the boundary between catheter and tissue indicated by a dark ring and the tissue visible in the centre of the image.

During the study procedure, the device is able to successfully image the septum walls, as demonstrated in Figure 6b. The transseptal cross is performed and the catheter is successfully navigated through it into the left atrium where it can continue with image acquisition, as shown in Figure 6d. The device is navigated further through the valve into the left ventricle where it is able to acquire further images around the apex area (Figure 6f).

## 6. Discussions

One of the primary issues were thin features on the parts breaking easily due to their limited structural thickness. At these small dimensions, through-holes can be challenging, especially over longer parts where the through-holes can fill in, requiring delicate post processing, or can present uneven internal surfaces, which made it difficult to pass components such as optical fibres through. In the final design iteration, the insertion into the sheathing lumen is improved by shaping the support with a smaller and filleted back tail, which also improves the attachment of the flush channel. The taper allows for more secure attachment of the fibres to the back of the piece without impinging the insertion into the catheter. In addition to assembly considerations, designs were heavily influenced by clinician feedback. Two in vivo studies were conducted using the devices, and the increase in the flush channel diameter as well as switching from one sheathing material to another were all influenced by lessons learned from several runs of experiments. The saline delivery had to be improved to deliver more volume. The earlier designs also had a lower resistance to kinking of the catheter while being pushed into the cardiovascular system through the guiding lumen, with revisions dictating a new material choice for the outer lumen. This in turn affected the dimensions of the distal alignment holder.

Since the device is navigated through the curving structure of the cardiovascular system, it must be designed with this requirement in mind. One important consideration is the material choice for the outer sheathing. The guiding catheter delivers the device to the location; however, the material must have both the flexibility to allow for easy navigation around bends, the rigidity to allow for it to be pushed through the catheter, and the kink resistance to prevent it from becoming stuck. This was a problem encountered in the first study use of the device and was revised after feedback from the clinicians. A Pebax sleeve was used (Nordson Medical) in favour of the braided polyimide initially used. The thicker wall of the new tubing also increased the whole stiffness of the catheter assembly while still maintaining the desired very good flexibility. The push ability issue was satisfactorily resolved by the material change.

Optical fibres, especially glass optical fibres, can negatively impact the stability of the device as it navigates bends. Figure 7 shows that light delivery is not affected dramatically even through tight bends. Losses through bending the fibres are minimal and not critical if enough light is delivered to the tissue and reflected back into the camera sensor. The purple Pebax material recovers well even after bending. However, breaking fibres is a common occurrence with the forces exerted on the devices. Hence, it may be desirable to switch to fibre bundles, or plastic optical fibres, in future design iterations. Note that, when experimenting on the use of polymeric fibres, a high intensity UV curing lamp has been found to melt the plastic, and different adhesion methods such as cyanoacrylate glue may have to be used and the assembly workflow adjusted slightly. These devices are single-use prototype devices for animal studies and, as such, stringent sterilisation requirements have not been taken into consideration, which may require some revision of the design [28].

Smaller integration was also explored and achieved by opting for a smaller camera sensor (OV6948), omitting the flush channel, and using only one illumination fibre (Figure 8). Here, a similar workflow to the one above is employed. We were able to achieve a footprint of 1.20 mm outer diameter using braided polyimide tubing. While such a system was not preferred for this cardiovascular application, due to the absence of the flush and inability to perform DRS measurements due to only one illumination fibre, it does demonstrate the feasibility of reducing the dimensions even further. In this instance, assembly was performed manually and without a distal alignment holder. This was due to the 3D printing limitation at the time, making a 3D printed stable holder difficult to source. Instead, the fibre and camera were carefully aligned and potted, before being drawn back into the lumen using a similar pull-back technique, and gaps were filled using UV cure adhesive. The adhesive was deposited using a micro-deposition tool and cured in stages.

## 7. Conclusions

We report here the design and assembly process of a chip-on-tip catheter that was used to successfully image in vivo cardiac tissues thanks to a microcamera sensor, an optical fibre-based illumination system, and the possibility to clear the field of view using a flush channel situated at the camera lens. We present design considerations and provide detailed assembly instructions for this device. These design and fabrication methods are transferable to other miniature catheter-based applications. The 3D printed support allows for adequate alignment of parts and packaging of the system in a 2.1 mm diameter catheter. It can be used as an adjunct tool for guidance in many different cardiac procedures. Images collected directly inside the heart could provide the clinicians and surgeons with vital information on the procedure and inform on the success of this as well. Further diameter reduction of the catheter footprint to reach smaller features could be possible by using more miniaturised imaging systems like the OmniVision OV6948 sensor, and integration at a 1.19 mm footprint was demonstrated, though omission of the flush channel in this system may require additional engineering.

The assembled catheter system with the flushing channel was used to capture images in an in vivo study on porcine models, which are, to the best of our knowledge, the first images captured using a microcamera sensor inside of a beating heart.

## Figures and Tables

**Figure 1 sensors-24-06216-f001:**
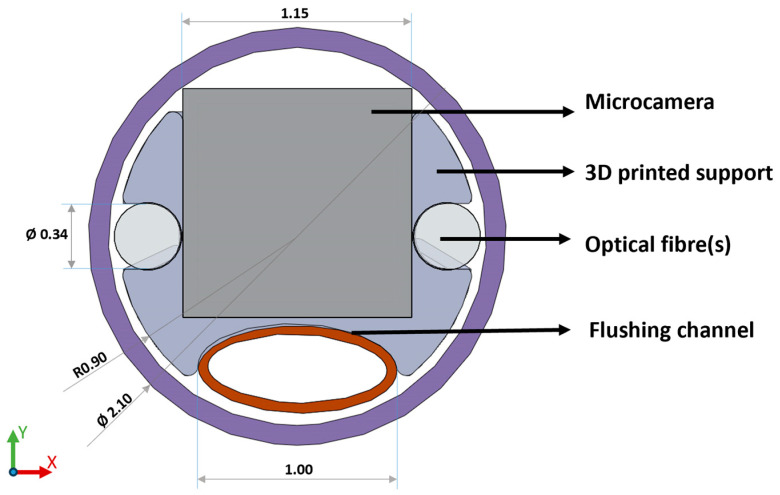
Front view of the final distal alignment holder design which aligns camera, optical fibres, and flush channel using a 3D printed holder which can be press fit and glued into the outer sheathing lumen. (Please note that the diameter dimension of the fibres used is 200 µm, while the fibre slot was printed with a diameter dimension of 340 µm to allow for glueing and material shrinkage. The same consideration applies to the camera slot, printed 1.15 mm, real dimension 1.10 mm.).

**Figure 2 sensors-24-06216-f002:**
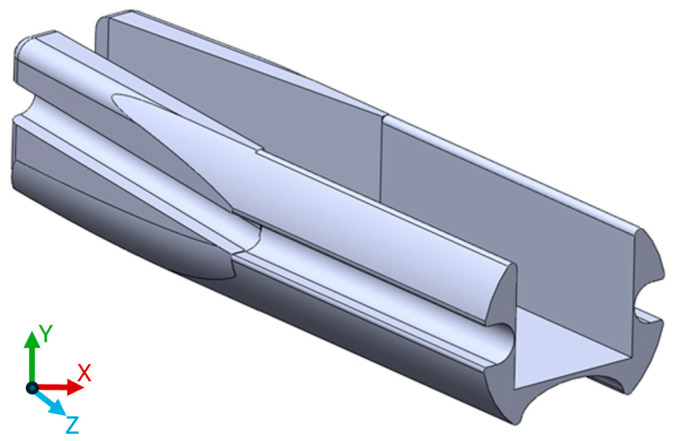
3D printed model used during this study. The length of the part on the z axis is 5 mm.

**Figure 3 sensors-24-06216-f003:**
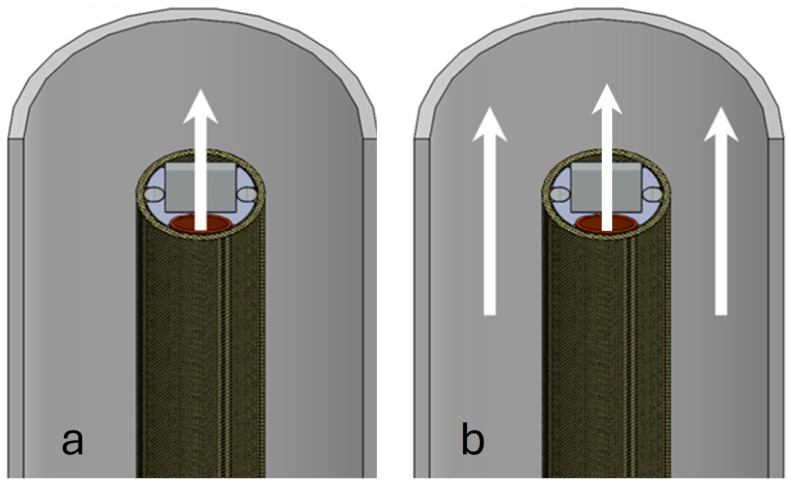
Schematic of blood field clearing methods. (**a**) shows the flow of saline flush coming from the channel in the catheter carrying the microcamera. In (**b**), the combined action of the flushing channel and the saline solution coming from the guiding catheter is shown.

**Figure 4 sensors-24-06216-f004:**
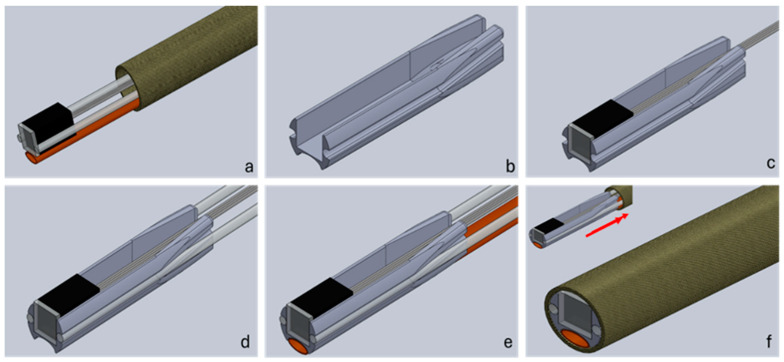
Assembly workflow of a distal end catheter-based imaging system. Cell (**a**) shows the components threaded through the sheathing lumen; (**b**) shows the 3D printed alignment holder. In step (**c**), the camera is inserted into the holder, followed by the fibres in step (**d**) and the flush channel in step (**e**). Finally, in step (**f**), the distal part is pulled back into the sheathing lumen and secured in place.

**Figure 5 sensors-24-06216-f005:**
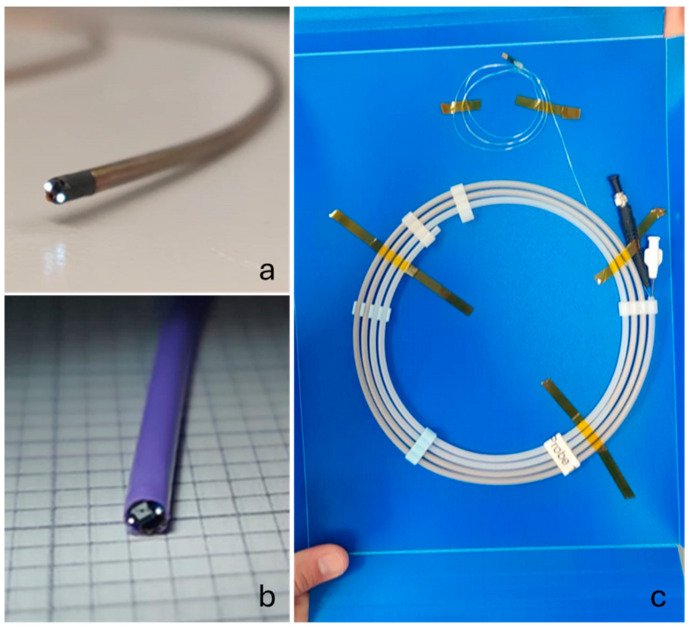
Finished 2.10 mm OD devices with the first iteration using braided polyimide tubing in (**a**), the final iteration with the Pebax tubing in (**b**), and the device securely packaged in a protective sleeve with the proximal terminations for the fibre, camera, and flush visible in (**c**).

**Figure 6 sensors-24-06216-f006:**
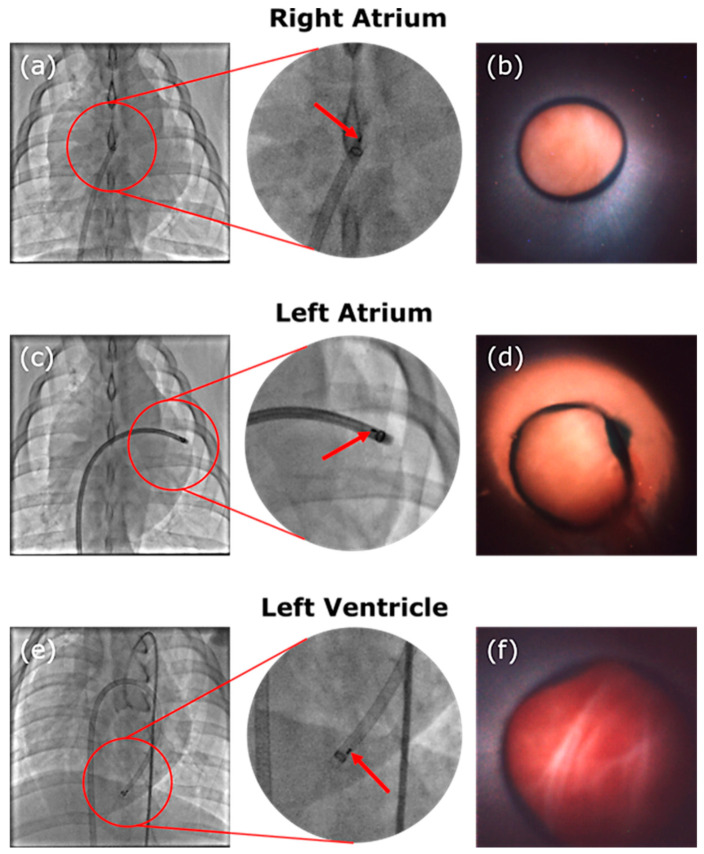
Images of the fluoroscopy showing the guiding catheter and device at location and corresponding camera images. The red arrow indicates the platinum–iridium marker bands attached to the distal end of the device. A circular marker band at the tip of the guiding catheter is also visible. The device was navigated to the right atrium in (**a**) and images the intra-atrial septum in (**b**). The device sits in the left atrium in (**c**), where an image of the endocardium in the left atrium is displayed in (**d**). Finally, the device is located in the left ventricle with its position visible in (**e**) and an image of the endocardium of the apex of the heart can be seen in (**f**).

**Figure 7 sensors-24-06216-f007:**
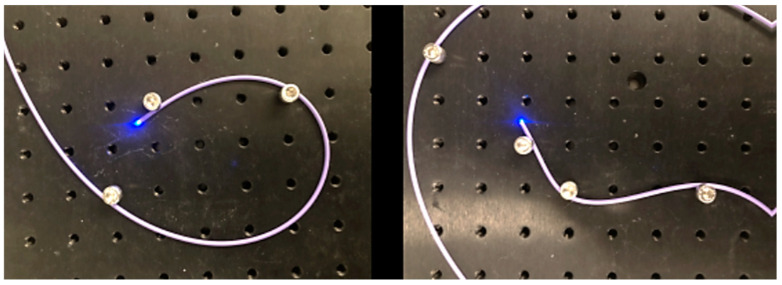
Light is delivered distally even through tight and complex bends.

**Figure 8 sensors-24-06216-f008:**
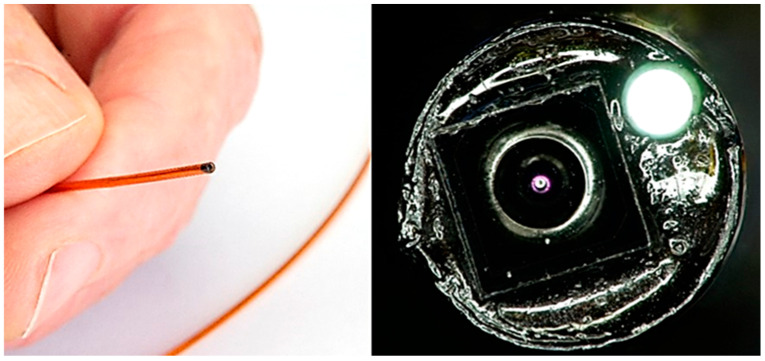
Assembled 1.20 mm outer diameter imaging catheter using OV6948 sensor, with illumination fibre. Shown held in hand for scale, alongside close-up microscope imaging, showing camera sensor and illuminated optical fibre.

## Data Availability

The original contributions presented in the study are included in the article, further inquiries can be directed to the corresponding author.

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
