# Peer review of "Design and Assembly of a Miniature Catheter Imaging System for In Vivo Heart Endoscopic Imaging"

_sensors, 2024, doi:10.3390/s24196216_

Round 1
Reviewer 1 Report
Comments and Suggestions for Authors
The text is raw and not suitable for scientific publication. Currently it looks like an assembly documentation. Before making any positive the decision concerning publication in Sensors the text should be entirely rewritten.
1. Scientific novelty is not presented. The scientific result is unclear. Comparison with the previously obtained results is not presented.
2. The structure of the paper should be given in the traditional form: Introduction, Materials and Methods, Results, Discussion, Conclusion. The sections are presented in such a way that it is difficult to separate the description of methods and results.
3. The article suffers from the description of a large number of different approaches that were considered during the research process but ultimately not applied. The description of the wrong steps taken by the researchers to achieve the research goal clutters the text and makes it difficult to understand the essence of the work. If the authors would like to present an advantage of the chosen approach over others, they should present a comparative study or at least give a comprehensive explanation of their choice. Depending on the statement the following parts should be eliminated or moved in the Discussion section in a reduced form or a comparative study to prove the statement should be presented.
Lines 83-85: “Initially, we used a catheter tubing with a fluoroscopy opaque band at the distal end. However, since the manufacturer (Creganna Medical, Galway, Ireland) had discontinued this product,”
Lines 98-104: “The printed parts were designed to be larger than the actual dimensions of the parts to account for shrinkage in the printing process. As per normal technological limitations of the printer and materials, some small details could not be achieved. Several iterations were carried out to finalise the design and the most suitable off-the-shelf material for the support. Some manual post processing of the parts was required to eliminate some unwanted resin residues and small stubs originated from the supporting structures created by the printer setup.”
Lines 110-115: “A small selection of the design iterations is shown in Figure 2. The early experimental layouts served to test the limitation of the 3D printing technology used, and to investigate materials. One of the early designs also included a prototype for a side facing imaging system. This design is possible and may have advantages depending on the intended application however was discarded for this study. All design iterations had similar assembling processes, varying slightly from one model to the other.”
Figure 2: leave only actual model. Indicate sizes. Also it can be combined with the Figure 1.
The description of the various approaches to flushing in subsection 3.2 appears unclear, scattered throughout the text, and contains ungrounded statements. This subsection should be rewritten entirely.
4. Line 162. “Each prototype device was assembled manually in-house.” How many prototypes are presented in the paper? It seems that only one was included in the study. If so, the description of only one prototype should be given.
5. Section 4. Assembly methods should be reduced to a couple of sentences carrying information about types of glues and X-ray markers. Currently it contains obvious or unnecessary information like “Due to the small dimensions, much of the assembly is performed under a microscope”. “The adhesive remains a liquid until cured and so allows for manipulation until it is cured”. “Placing the glue is performed manually. Small amounts of the glue can be picked up and placed using a micro-applicator. This can also be sharp instrument such as a needle, or if scratching the optical components is a concern, a sharpened piece of wood, such as a whittled toothpick.”, Figure 3c showing secure packaging, etc.
Minor
1. Line 30: [6] instead of .6
2. Line 55: micro-image instead of micro-i2mage
3. Line 337: it seems it is Conclusion, not Discussion section
4. Lines 89-91: looks like editorial notes from the PI. Please, delete it
5. XY axes on the Figure 1 would be beneficial
6. Resolution of all Figures should be increased
Author Response
Please see the attachment, thank you

Reviewer 2 Report
Comments and Suggestions for Authors
In this work, the authors introduce the procedure of the miniature catheter imaging systems based on miniature cameras. The authors also demonstrated imaging of the heart of pig animal model. While the direction of the manuscript falls within the category of the journal, I would like to suggest the authors to modify and address the comments and concerns below.
1. I would like to suggest the type of the manuscript would be changed to ‘Technical note’ or ‘Project report’. I do not think this study can be considered a research article due to the way of explanation and the completeness of the work.
2. The authors may be able to introduce relevant works that used miniature cameras to further elaborate the novelty of the work.
3. Probably, one of the corresponding authors are not adequately designated.
4. Ref. 6 is not with parentheses (line 30).
5. 2-3 mm on line 45 should be 2–3 mm (hyphen should be replaced with en dash).
6. ‘In vivo’ is usually in Italic.
7. mm on line 69 should be mm3. Alternatively, 1.10 mm×1.10 mm×1.50 mm is valid, too.
8. On line 72, um should be µm.
9. What is the focal length and the field of view of the camera?
10. A sentence on lines 89–91 may be discarded.
11. Number and unit should have a space in between (e.g., line 58, 2.3mm=>2.3 mm).
12. What 3D printer did you use in the study?
13. Please add the size and scale on Fig. 1. Please designate another optical fiber, too.
14. What was the reason to discard the design mentioned on lines 112–114.
15. The image quality of the manuscript is not good enough for journal publication.
16. The sentence on lines 233–236 seems strange. Please modify and proofread the paragraph in Section 5.
17. In Fig. 6, the parts imaged seem different from the description (Section 5 as well). Please double-check it.
18. Could the authors show the images of the different part of the heard without using flushing mechanism?
19. Could the authors perform comparison of the heart images with a traditional endoscopy?
20. The authors mentioned that the system is capable of DOS measurement. Could the authors show DOS signals acquired using the system?
21. Could the authors perform quantitative measurement of the optical power for the experiment of Fig. 7?
Comments on the Quality of English Language
I mentioned it in the previous comment section.
Author Response
Please see the attachment, thank you

Round 2
Reviewer 1 Report
Comments and Suggestions for Authors
It was supposed that the Authors will cordinally revise the manuscript to stress the novelty and the scientific merit of the research, but instead of this they only answered my questions in the cover latter and didn't make appropriate corrections in the manuscript. That's why, unfortunately, I can't recommend this text for publication in the present form.
Author Response
Dear Reviewer 1,
Thank you again for your reply to this manuscript. We understand your point and we have modified the abstract and the Study Aim (along the lines in our comments sent back to you earlier) highlighting the novelty of this research. We are now explaining better what the aim of this paper is and why we think it could be beneficial to the research community. Furthermore, we have highlighted the use of the saline flush as well in the abstract and we have included that it is “novel” when describing the system.
We had also already included in the previous version of the manuscript that this is the first time in literature that endoscopic images in-vivo have been captured in a beating heart.
We have also substantially modified the manuscript, as suggested in the previous round of revisions, by removing and modifying several parts of the text body.
Attached you will also find a comparison of the original document with the latest reviewed version of the manuscript.
Thank you

Reviewer 2 Report
Comments and Suggestions for Authors
Just a minor comment. I would like to recommend the authors to change the title as "Design and assembly of a miniature catheter imaging system for in-vivo heart endoscopic imaging"
Comments on the Quality of English LanguageThe sentence in lines 212-214 is still gramatically wrong.
Author Response
Dear Reviewer 2,
Thank you again for your reply to this manuscript. We have modified the title as suggested and we will communicate this to the editor to have this corrected in the final version of the manuscript.
We understand how that particular sentence is difficult to understand in the way it was presented. We have changed it again and hopefully this time will make more sense grammatically.